# Effect of Combined Protein-Enriched Enteral Nutrition and Early Cycle Ergometry in Mechanically Ventilated Critically Ill Patients—A Pilot Study

**DOI:** 10.3390/nu14081589

**Published:** 2022-04-12

**Authors:** Ilya Kagan, Jonathan Cohen, Itai Bendavid, Sandy Kramer, Ronit Mesilati-Stahy, Yehuda Glass, Miriam Theilla, Pierre Singer

**Affiliations:** 1Department of General Intensive Care and Institute for Nutrition Research, Rabin, Medical Center, Beilinson Campus, Petah Tikva 49100, Israel; jdcspc@gmail.com (J.C.); itaibd@clalit.org.il (I.B.); sandyk@clalit.org.il (S.K.); theillamiriam@gmail.com (M.T.); psinger@clalit.org.il (P.S.); 2Laboratory of Nutrition and Metabolism Research, Felsenstein Medical Research Center, Sackler Faculty of Medicine, Tel-Aviv University, Rabin Medical Center, Beilinson Campus, Petah Tikva 49100, Israel; ronit_mesilati@walla.com; 3Medical Intensive Care Unit, Rambam Health Care Campus, Technion Faculty of Medicine, Haifa 31096, Israel; ydglass@gmail.com; 4Steyer School of Health Professions, Nursing Department, Sackler School of Medicine, Tel Aviv University, Tel Aviv 69978, Israel

**Keywords:** critical care, enteral nutrition, cycle ergometry, high-protein diet, physiotherapy

## Abstract

Background: Cycle ergometry (CE) is a method of exercise used in clinical practice. Limited data demonstrate its effectiveness in critically ill patients. We aimed to evaluate the combination of CE and a high-protein diet in critically ill patients. Methods: This was an open label pilot trial comparing conventional physiotherapy with enteral nutrition (EN) (control, Group 1), CE with EN (Group 2), and CE with protein-enriched EN (Group 3). The primary outcome was length of ventilation (LOV). Secondary outcomes were intensive care unit (ICU) mortality, length of ICU stay (ICU LOS), length of hospital stay (Hospital LOS), and rate of re-intubation. Results: Per protocol, 41 ICU patients were enrolled. Thirteen patients were randomized to Group 1 (control), fourteen patients to Group 2, and fourteen patients to Group 3 (study groups). We found no statistically significant difference in LOV between the study arms (14.2 ± 9.6 days, 15.8 ± 7.1 days, and 14.9 ± 9.4 days, respectively, *p* = 0.89). Secondary outcomes did not demonstrate any significant differences between arms. Conclusions: In this pilot trial, CE combined with either standard EN or protein-enriched EN was not associated with better clinical outcomes, as compared to conventional physiotherapy with standard EN. Larger trials are needed in order to further evaluate this combination.

## 1. Introduction 

The incidence of Intensive Care Unit Acquired Weakness (ICUAW) is reported as being 25–60% after one week of mechanical ventilation and may impair the long-term outcome of ICU patients (such as morbidity and mortality, decreased cognitive and functional status upon discharge, and decreased quality of life [1,2]). Most of these functional impairments can persist for up to 5 years after discharge from ICU [3].

Early mobilization with standard physical therapy may help to reduce the effects of ICUAW by increasing muscle strength, reducing days of mechanical ventilation, decreasing LOS in the ICU, and increasing in-hospital function [4]. This has been shown to be safe and feasible [5,6]. Cycle ergometry is one of the new methods that has been suggested in clinical practice over the last few decades. It is a stationary bicycle that can be used in the supine or sitting position, and can help mobilize the patient through either passive or active cycling. It can also be used with patients on mechanical ventilation [7,8].

Biolo et al. suggested that in normal volunteers, the association of exercise and protein intake improved protein synthesis [9]. We were not aware of studies evaluating the combination of a high-protein diet and early physiotherapy using cycle ergometry in ICU patients. Therefore, we conducted a pilot randomized control trial, comparing the effect of conventional physiotherapy with enteral nutrition (EN), cycle ergometry with EN, and cycle ergometry with protein-enriched EN on the duration of ventilation in ICU patients.

## 2. Methods

### 2.1. Study Design

This was a single-center, prospective, open label, randomized controlled trial. The study was performed in the general ICU of Rabin Medical Center, Beilinson Hospital, Israel, a tertiary care, Level 1 Trauma Center of a university-affiliated hospital, over a period of 4 years (from 1 January 2013 to 31 December 2016). The study protocol was approved by the local institutional review board, and informed consent was obtained prior to randomization, either from patients, their legal representative (if possible), or an independent physician where this was not possible. The study was registered at Clinical Trials as NCT01099501.

### 2.2. Participants

Adult patients (age 18–90 years) who had been mechanically ventilated for at least 48 h with an expected period of ventilation of a minimum of 7 days were eligible for inclusion. Exclusion criteria were: (a) conditions that impaired the cycling movement; (b) trauma, arthritis, or surgery of the leg, pelvis, or lumbar spine; (c) open abdominal wounds or abdominal compartment syndrome; (d) an anticipated fatal outcome of ICU; (e) pre-existing diagnosis of neuromuscular weakness, acute stroke, or status epilepticus; (f) cardiorespiratory instability (need for significantvasopressor support (noradrenaline) >0.2 mcg/kg/min, dopamine >8 mcg/kg/min, inspiratory oxygen fraction (FIO_2_) up to 60% or PEEP (positive end expiratory pressure) up to 10 cm H_2_O or treatment by NO (nitric oxide), minute ventilation 150 mL/kg body weight, respiratory rate 30 breaths/min on adequate ventilatory support); (g) contra-indication for EN, including mechanical or functional bowel obstruction, high output fistula, severe necrotizing pancreatitis; (h) pregnancy.

### 2.3. Randomization and Allocation

Eligible patients were randomized in a 1:1:1 ratio into 3 groups: conventional physiotherapy with EN (control, Group 1), cycle ergometry with standard EN (Group 2), and cycle ergometry with protein-enriched EN (Group 3). Randomization was achieved using a computer-based block randomization generated by a statistical software program, which was concealed to all investigators apart from the principal investigator (PI). Investigators and clinicians were unblinded to the treatment allocation.

### 2.4. Interventions

Group 1 was treated using conventional physiotherapy and received EN (Jevity^®^, Abbott, Chicago, IL, USA) according to the individual energy requirements assessed by the indirect calorimetry measurement of Resting Energy Expenditure (REE). Patients in this group were treated by conventional (respiratory) physiotherapy adjusted to the individual needs and a standardized mobilization session of the upper and lower extremities, 5 days per week. Passive motion was applied in sedated subjects, whereas awake patients were asked to participate actively. The intensity of the exercises was increased according to the patient’s capability. Each session was continued for a minimum of 20 min. Groups 2 and 3 were randomized to receive EN (Jevity^®^, Abbott) and protein-enriched EN (very-high-protein formula Promote^®^, Abbott), respectively, according to the individual energy requirements assessed by the indirect calorimetry measurement of REE, similar to the control group. Groups 2 and 3 were treated using cycle ergometry (MOTOmed viva2, Medimotion, Carmarthenshire, Wales, United Kingdom, SA39 9AZ). Quadricep muscle strength was evaluated in conscious patients using manual muscle testing, where the individual was asked to hold a limb or other body part at the end of its available range, or at another point in its range of motion, while the clinician provided manual resistance. The active passive trainer (cycling ergometry) was used to mobilize and strengthen the subjects. When patients were able to cycle actively, the cycling session was divided into two bouts of 10 min or into more intervals when needed. At the start, after 10 min and the end of every session, training intensity was evaluated and vital signs were recorded. Patients were placed in a comfortable position in between the supine and the semi recumbent position. For sedated patients, cycling was performed in a passive manner for 20 consecutive minutes.

Conventional physiotherapy or cycle ergometry were started during the first 24 h after randomization. EN, based on randomization, was delivered immediately via a nasogastric or orogastric tube whose position was confirmed by X-ray.

The amount of EN prescribed daily was meant to provide at least 80% of all energy requirements as determined by the measurement of resting energy expenditure (REE) (Deltatrac II; Datex-Ohmeda, GE Healthcare Finland, Helsinki, Finland). REE measurements were performed before study inclusion (in the first 48 h after admission) and at intervals of 48–72 h thereafter. REE was calculated using Weir’s formula [10]. Patients were measured in stable state, without any changes in their respiratory or hemodynamic status in the last 30 min before measurement. Patients with an FIO_2_ above 0.6 were not measured. If the measured REE was not applicable, the Fagon formula was used [11]. Tolerance to EN was assessed by measuring gastric residual volume (GRV) every 8 h. In the presence of a GRV of more than 500 mL and/or vomiting or diarrhea 3 times/day, EN was stopped for 24 h. During this period, CE or physiotherapy were continued and study was not stopped. If the GRV was between 150 and 500 mL, the EN rate was reduced to 50% and/or prokinetic therapy (metoclopramide at 10 mg^3^/day or erythromycin at 80 mg^3^/day) was initiated. If the GRV remained 500 mL and 80% of caloric needs were met by EN alone, the feeding regimen was continued.

The study’s intervention was continued until ICU discharge, death, or the completion of 28 days of the study, whichever came first.

### 2.5. Safety

Safety assessment included patient comfort (pain), an increase in heart rate of more than 15% from the baseline, an increase or decrease in systolic blood pressure (SBP) of 15%, a decrease in SpO_2_ below 92%, the occurrence of signs of pulmonary edema or coronary syndrome, or a decrease in pH below 7.15 during CE.

### 2.6. Screening and Baseline Measurements

Baseline characteristics, including age, sex, weight, height, body mass index, admission diagnosis, Acute Physiology and Chronic Health Evaluation II (APACHE II) score, and Sequential Organ Failure Assessment (SOFA) score, were recorded at enrollment.

### 2.7. Study Outcomes

Primary outcome: The primary outcome was duration of mechanical ventilation. Successful weaning was defined if the patient was breathing for 48 h without invasive support.

Secondary outcomes: Secondary outcomes included: (a) ICU mortality, (b) length of ICU stay, (c) length of hospital stay, and (d) re-intubation rate.

### 2.8. Statistical Analysis

All statistical analyses were performed for an intention to treat (ITT) and a per protocol (PP) population. Patients who received mechanical ventilation for less than 48 h or who were intolerant of EN for more than 3 days were excluded from per protocol analysis. Data are presented as mean values ± standard deviation (SD). Statistical analyses of clinical and basic parameters were conducted with repeated-measures analysis of variance (ANOVA) with the SPSS version 12.0 software (SPSS Inc., Chicago, IL, USA) for Windows. The correlations were evaluated using the chi-square test.

### 2.9. Sample Size Estimation

Power analysis was conducted using software (G*Power 3 for Wind, Heinrich Heine University, Dusseldorf, Germany) during the planning of this study. The effect size was estimated from the PEP uP study which compared two groups of 20 and 30 patients [12]. The detection effect amount was set as SD × 1, with the level of significance of 0.05, and power of 0.8. Therefore, the necessary sample size for each group was calculated as 25.

## 3. Results

From January 2013 to December 2016, a total of 85 patients were screened, and 62 of them were enrolled in the study. Forty-one patients completed follow up. A study flow chart of the recruitment process is shown in Figure 1. The study was stopped early due to the slow recruitment rate. Finally, 62 patients were included in the intention to treat analysis. Of those 62 patients, 22 were excluded (13 due to early extubation, and 6 due to early unplanned discharge or transfer to another department, prior to completing 7 days of mechanical ventilation in ICU, or who died during the first 7 days of ventilation), and 2 due to their inability to properly tolerate EN (TPN (total parenteral nutrition) for more of 3 days). Thus, the per protocol analysis included 41 patients, of whom 13 formed the control group and 28 formed interventional groups (14 patients in each group).

Baseline characteristics of included patients are presented in Table 1. There were no significant differences between the three groups at baseline. 

## 4. Outcomes

### 4.1. Primary Outcome

Formulas compositions and nutrients data are represented in Table 2. The total energy and protein intake and fluid balance are represented in Table 3. There was no significant difference between groups regarding LOV in both the ITT and per protocol analyses. In the ITT population (Groups 1, 2, and 3), mean LOV was 10.2 days (±SD 9.5), 12.0 days (±SD 7.8), and 11.7 days (±SD 9.7), respectively (*p* = 0.79). In the per protocol population, the mean LOV in the control group was 14.2 days (SD ± 9.6, 13 patients), in Group 2 it was 15.8 days (SD ± 7.1, 14 patients), and in Group it was 3–14.9 days (SD ± 9.4, 14 patients) (*p* = 0.89).

### 4.2. Secondary Outcomes

No significant differences between the study arms were demonstrated for any of the secondary outcomes, including ICU and hospital length of stay, and ICU and hospital mortality in either the PP or the ITT population (Table 2). Successful weaning was documented in 10/22 (45.5%) in Group 1, 7/21 (33.3%) in Group 2 and 9/19 (47.4%) in Group 3, without statistically significant differences between groups (*p* = 0.61) in the ITT population, and 4/13 (30.8%) in Group 1, 4/14 (28.6%) in Group 2, and 6/14 (42.9%) in Group 3 (*p* = 0.69) in the PP population.

## 5. Discussion

In this pilot randomized, open label, single-center study, we compared patients who received a high-protein-concentration diet and early exercise, versus patients who were treated with a regular diet. No significant differences between these groups were found for the primary outcome of LOV in ICU. Secondary outcomes, including rate of weaning, hospital and ICU length of stay, and hospital and ICU mortality were also without significant differences.

To the best of our knowledge, this is one of the first pilot randomized controlled trials evaluating the combination of high protein concentration with early cycle ergometry, for critically ill patients.

Previous studies investigated protein intake and early physical activity separately. Sarcopenia is a common condition in ICU that is correlated with mortality and morbidity in the ICU. Peterson et al. [13] found that an incidence of sarcopenia in ICU correlates with a 60–70% chance of a worse outcome (mortality and length of ICU stay), especially in older patients. Woo et al. [14] showed that poor skeletal muscle mass was associated with extubation failure after prolonged mechanical ventilation, underlining the importance of the skeletal muscle mass in ventilated patients. In a single-center cohort study, Weijs et al. [15], showed improvement in 3 months mortality after ICU discharge in surviving patients who had better protein balance. In a retrospective cohort study, our group [16] found an improvement in ICU survival associated with early protein administration in a mixed ICU department. High doses of protein have been recommended by the latest ESPEN guidelines of nutrition in ICU, together with physical activity [17]. However, the effects of increased protein administration during the acute and post-acute phase are still a matter of debate. While some authors [18] have pointed out an association between early administration of protein and decreased ICU discharge rate, or an increase in fat accumulation in the biopsies of patients receiving early protein administration [19], others found an improvement in protein synthesis 30 days after injury [20]. ESPEN guidelines that were published after the initiation of our study recommend 1.3 g/kg/d of protein at an early stage, and definitely after 7 days. Our study suggested to provide standard formula in the two first groups (control and with CE), and a protein-enriched formula in the third group (14.8 g versus 10.4 g per 235 mL of formula), without increasing the energy load. An increase in protein using this formula would have increased the energy administration and risked overfeeding.

Over the past 10 years, early physical activity for critically ill patients has been investigated in several studies. The European Respiratory Society and European Society of Intensive Care Medicine recommended that intubated patients should receive early active or passive mobilization and muscle training [21]. Many patients who were treated by mechanical ventilation did not receive early mobilization, and more than 50% of them suffered from ICU-acquired weakness [1]. Burtin et al. [22] investigated daily physical activity by using a bedside CE for the prevention of deterioration of exercise capacity. No information was reported in that study regarding nutrition. Therefore, the improvement of self-perceived functional status and muscle force at hospital discharge could not be related to nutrition. Schaller et al. [23] showed an improvement in length of stay and functional mobility in surgical patients who received early mobilization. In another study [24], an early protocol-based physiotherapy and muscle activation with neuromuscular electrostimulation did not improve muscle strength or function, but did prevent muscle atrophy. Fossat et al. [25] compared two large groups of patients receiving standard physical to in-bed cycling and electrical stimulation of the quadriceps on global muscle strength. The authors did not find a significant difference in the ICU mobility scale score, and the median number of ventilator-free days at day 28 was 21 (IQR, 6 to 25) in the intervention group, compared to 22 (IQR, 10 to 25) in the usual care group (median difference, 1 (95% CI, −2 to 3); *p* = 0.24). We initiated our study earlier (2013) when electrical stimulation was not used routinely and decided to only add one session of in-bed leg cycling, in addition to increased protein administration, assuming that the association of high protein intake to physical activity would improve clinical outcomes. Others [24] also showed that muscle activating measures did not improve muscle strength or function at first awakening, but prevented atrophy.

Finger et al. [26] showed that protein supplementation with resistance training of muscle mass was associated with an increase in free fat mass, but was not associated with elevated muscle mass and strength. Nakamura et al. [27] compared patients who received an energy load (20 kcal/kg/d with either 1.5 g/kg/d or 0.8 g/kg/d of protein) together with rehabilitation using a belt-type electrical muscle stimulation, once a day for 20 min. The association of high protein and electrical muscle stimulation provided a better muscle volume. However, the length of ventilation, length of stay, and survival were not different between the two groups, similar to our study. Nosocomial pneumonia occurrence was lower in the high-protein group, but not significantly.

Recently, the addition of β-Hydroxy-β-methylbutyrate (HMB) has been suggested to improve protein synthesis in critically ill patients [28]. However, another prospective randomized trial [29] administrating HMB during the acute phase did not inhibit the muscle volume loss of patients with severe multiple trauma.

Our results are disappointing but have been confirmed by others. Blanc-Bisson et al. [30] prescribed intensive and early physiotherapy in acutely ill elderly patients together with improved dietary intake, but could not find any advantages. No study that assessed associating higher protein intake with early physical activity succeeded in improving short-term clinical outcomes at the time this study was started. Recently, a study administrating high doses of protein with early exercise [31] followed the long-term outcomes of patients and found a significant improvement in physical component summary at 3 and 6 months, as well as a significant increased survival, suggesting that short-term outcomes may not be the optimal objectives to study in this approach.

Our study has some limitations. The main limitation was the sample size obtained in the per protocol analysis. Therefore, the impact of our pilot study is limited. This sample size was smaller than that used in the power analysis. This fact makes our findings less significant. The difficulty with patient recruitment was the result of a broad spectrum of contra-indications that we decided in our study design. These facts should guide the design of further studies. The recruitment rate was slow and the study was continued for more than 3 years. The per protocol groups of patients were small due to the exclusion of the patients during the study, mainly in relation to early and unexpected extubation. The implementation of an adequate rehabilitation program to prevent physical decline, hospital re-admission, and even institutionalization after acute illness supposes availability of skilled physiotherapists in intensive care units. In addition, this study was performed in one center. In our study, we were limited by physiotherapists’ availability and we were unable to administer more than 20 min per day of physical activity. In addition, despite planning this procedure in the protocol, we failed to check handgrip in our patients because most of them were unable to perform this test in the first days of the study period. Finally, our study only looked at weaning success and not at long-term survival or function.

## 6. Conclusions

In this pilot study, performing early physiotherapy alone or in combination with cycle ergometry and protein-enriched diet was not associated with an improved length of ventilation and or weaning rate. These results may be related to low recruitment. Larger randomized studies will be required to determine the best time for starting physiotherapy and the number of required sessions, as well as to define the adequate protein amount to administer in order to improve the clinical outcome of ICU patients.

## Figures and Tables

**Figure 1 nutrients-14-01589-f001:**
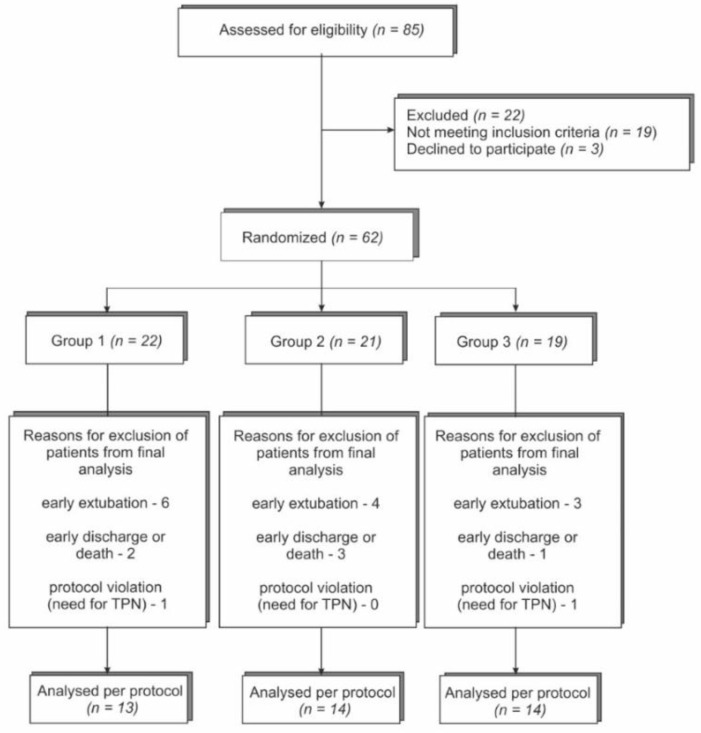
Study flow chart of patient enrollment; TPN—total parenteral nutrition.

**Table 1 nutrients-14-01589-t001:** Baseline and clinical characteristics of the three groups.

Parameters	ITT Group (62)				Per-Protocol Group (41)			
	Group 1 (*N* = 22)	Group 2 (*N* = 21)	Group 3 (*N* = 19)	*p*	Group 1 (*N* = 13)	Group 2 (*N* = 14)	Group 3 (*N* = 14)	*p*
Age	64 ± 13	63 ± 18	61 ± 16	NS	66 ± 14	57 ± 19	58 ± 15	NS
Male%	63.6	57.1	73.7	NS	61.5	57.1	71.4	NS
APACHE II	21 ± 6	20 ± 6	22 ± 8	NS	21 ± 6	21 ± 6	21 ± 8	NS
SOFA (admission)	6 ± 3	5 ± 2	7 ± 4	NS	5 ± 2	5 ± 2	7 ± 4	NS
BMI	29 ± 8	29 ± 5	30 ± 7	NS	32 ± 7	29 ± 6	30 ± 7	NS
Pre study LOV (days)	5 ± 4	8 ± 6	7 ± 5	NS	5 ± 4	8 ± 7	7 ± 4	NS
Mean REE (IC)	1836 ± 552	1876 ± 618	1820 ± 220	NS	1867 ± 509	1994 ± 728	1813 ± 251	NS
Diagnosis in admission				NS				NS
Pneumonia	6	7	4		4	4	4	
Sepsis	6	3	4		5	3	3	
COPD	0	3	0		0	2	0	
CHF	3	2	6		2	1	4	
Trauma	5	4	4		2	3	2	
ARDS	2	0	1		0	1	1	
Airway protection	0	2	0		0	0	0	
Mean Fagon	2070 ± 277	1974 ± 230	2022 ± 273	NS	2088 ± 313	1997 ± 534	1998 ± 261	NS
Pre study albumin	2.7 ± 0.4	2.4 ± 0.4	2.7 ± 0.5	NS	2.4 ± 0.3	2.4 ± 0.5	2.5 ± 0.5	NS
Mode of ventilation				NS				NS
A/C	2	2	1		1	2	1	
SIMV	15	11	13		8	6	9	
PSV	5	5	5		4	4	4	
ASV	0	3	0		0	2	0	

ITT—intention to treat, APACHE—acute physiology and chronic health evaluation, SOFA—sequential organ failure assessment, BMI—body mass index, LOV—length of ventilation, REE—rest energy expenditure, IC—indirect calorimetry, COPD—chronic obstructive pulmonary disease, CHF—congestive heart failure, ARDS—acute respiratory distress syndrome, A/C—assist control mechanical ventilation; SIMV—synchronizes intermittent mechanical ventilation; PSV—pressure support ventilation; ASV—adaptive support ventilation, *p*—*p* value, NS—not significant.

**Table 2 nutrients-14-01589-t002:** Composition of study and control formulae (amount per serving (237 mL)).

Characteristics and Nutrient Data	Study Formula (Promote)	Control Formula (Jevity)
Nutrient density, Cal/mL	1.0	1.06
Protein,% Cal	25	16.7
Carbohydrate, % Cal	50	54.3
Fat, % Cal	25	29.0
MCT/LCT	19:81	19:81
Protein, g	14.8	10.4
Carbohydrate, g	32.8	36.5
Dietary Fiber, g	3.4	3.4
Fat, g	6.7	8.2
Water g/mL	197	197
Energy, Cal	237	250

MCT/LCT—medium chain triglyceride/long chain triglyceride.

**Table 3 nutrients-14-01589-t003:** Primary and secondary outcomes.

Parameters	ITT Group (62)				Per-Protocol Group (41)			
	Group 1 (*N* = 22)	Group 2 (*N* = 21)	Group 3 (*N* = 19)	*p*	Group 1 (*N* = 13)	Group 2 (*N* = 14)	Group 3 (*N* = 14)	*p*
LOV in ICU	10.2 ± 9.5	12.0 ± 7.8	11.7 ± 9.7	NS	14.2 ± 9.6	15.8 ± 7.1	14.9 ± 9.4	NS
ICU LOS	17.2 ± 9.6	16.3 ± 7.7	18.8 ± 10.5	NS	19.9 ± 10.1	19.3 ± 7.2	20.8 ± 10.1	NS
Hospital LOS	33.1 ± 22.6	26.5 ± 17,0	35.2 ± 25.7	NS	40.8 ± 23.4	30.7 ± 17.9	36.2 ± 27.4	NS
Weaning	10 (45%)	7 (33.3%)	9 (47.4%)	NS	4 (30.8%)	4 (28.6%)	6 (42.9%)	NS
ICU mortality	1 (4.5%)	3 (14.3%)	3 (14.3%)	NS	1 (7.7%)	3 (21.4%)	3 (21.4%)	NS
In hospital mortality	4 (18.2%)	5 (23.8%)	5 (23.8%)	NS	4 (30.8%)	4 ((28.6)	5 (35.7%)	NS
Daily caloric intake	1557.6 ± 309.5	1648.2 ± 375.8	1372.7 ± 530.8	NS	1547.8 ± 239.2	1710.2 ± 324.9	1532.1 ± 323.9	NS
Daily protein intake	63.6 ± 13.6	67.2 ± 20.2	83.7 ± 31.9	0.02	61.6 ± 10.5	70.9 ± 22.7	83.5 ± 24.7	0.03
Fluid balance	2364 ± 9045	2013 ± 9289	3233 ± 10673	NS	4026 ± 11022	2898 ± 11074	3036 ± 12188	NS

ITT—intention to treat, LOV—length of ventilation, LOS—length of stay, ICU—intensive care unit, *p*—*p* value, NS—not significant.

## Data Availability

Data will be available only after agreement of our HMO.

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
