# Peer review of "Effect of Combined Protein-Enriched Enteral Nutrition and Early Cycle Ergometry in Mechanically Ventilated Critically Ill Patients—A Pilot Study"

_nutrients, 2022, doi:10.3390/nu14081589_

Round 1

Reviewer 1 Report

The aim of this manuscript is to evaluate the effect of “protein enriched enteral nutrition and early cycle ergometry” on mechanically ventilated patients. The primary outcome was duration of MV. The secondary outcomes were ICU mortality, length of ICU stay and re-intubation rate. This is a single center, RCT conducted in the period of 2013 to 2016.

Major concerns

  1. The most critical problem in the current study is the sample size is too small to make it sufficient to evaluate the treatment effect. In addition, there were three groups in the current study, which required more case numbers to do the between group analysis. In the per-protocol group, the case numbers in group 1, 2 and 3 were 13, 14 and 14, respectively. These small numbers even not match the minimal requirement as 25 cases in each group, according to sample size estimation in the methods. The results showed both primary and secondary outcomes were negative findings. Therefore, we don`t know whether the results are truly negative or just due to the insufficient of case numbers.

  1. The primary outcome is not a robust outcome. As you mentioned in the first paragraph of discussion, sarcopenia is thought to be the major problem of critically ill patients in ICU. Sarcopenia will contribute to ICU mortality, prolong respiratory failure and long-term disability. Therefore, the value of the current study will be to evaluate the effect of early cycle ergometry in combination of protein enrich feeding on improving sarcopenia in critically ill patients. In fact, ventilator days is not a robust outcome. In your study protocol, patients with expected period of MV for a minimum of 7 days were enrolled. Therefore, patient fit the criteria of extubation with MV support less than 7 days were excluded from the final analysis. As you showed in the figure 1, 13 of 62 (21%) enrolled patients were excluded due to this reason. This is another lethal point regarding the study design.

  1. In the current study, patients were classified into three groups. Group 1 is the control group; Group 2 is the cycle ergometry (CE) group; and Group 3 is combined cycle ergometry and protein enriched EN. I think if the sample size is enough, you can do the 2 x 2 comparison to compare the effect of CE and protein enriched EN. However, the case number is insufficient to do the analysis.

Minor concerns

  1. The quality of figure 1 is poor.
  2. References: several references miss in the manuscript. For example, reference 12 and 30.
  3. Blanc- Bisson et al will be the reference 32 rather than 30 as you shown in the discussion part.

Reviewer 2 Report

Cycle ergometry (CE) is a method of exercise using in clinical practice. But from the above, the assumption of a good protein intake and sarcopenia is severely limited in this work by the inconstancy and little dedication to (CE). Therefore a well-conducted work in terms of recruitment and quality control but limited to the certain observation that there is no correlation between protein intake and benefits in patients undergoing mechanical ventilation.

Round 2

Reviewer 2 Report

In the light of the revisions made it can be a valid scientific communication.

Author Response

We thank the reviewer for his comments stating that in the light of the revisions, this can be a valid scientific communiaction